# In vitro evidence against productive SARS-CoV-2 infection of human testicular cells: Bystander effects of infection mediate testicular injury

**Stefanos Giannakopoulos[1], Daniel P. Strange[2], Boonyanudh Jiyarom[2], Omar Abdelaal[3,4], Aaron W. Bradshaw[3,5], Vivek R. Nerurkar[2], Monika A. Ward[6], Jackson Bakse[6], Jonathan Yap[7], Selena Vanapruks[7], William A. Boisvert[7], Michelle D. Tallquist[7], Cecilia Shikuma[2], Hooman Sadri-Ardekani[3,5], Philip Clapp[3], Sean V. Murphy[3], Saguna Verma** [1,2] *

1 Department of Cell and Molecular Biology, John A. Burns School of Medicine, University of Hawaii at Manoa, Honolulu, Hawaii, United States of America, 2 Department of Tropical Medicine, Medical Microbiology, and Pharmacology, John A. Burns School of Medicine, University of Hawaii at Manoa, Honolulu, Hawaii, United States of America, 3 Wake Forest Institute for Regenerative Medicine, Wake Forest School of Medicine, Winston-Salem, North Carolina, United States of America, 4 Department of Urology, Faculty of Medicine, Zagazig University, Egypt, 5 Department of Urology, Wake Forest School of Medicine, Winston-Salem, North Carolina, United States of America, 6 Institute for Biogenesis Research, John A Burns School of Medicine, University of Hawaii at Manoa, Honolulu, Hawaii, United States of America, 7 Center for Cardiovascular Research, Department of Medicine, John A. Burns School of Medicine, University of Hawaii at Manoa, Honolulu, Hawaii, United States of America

* saguna@hawaii.edu

**Data Availability Statement:** All relevant data are within the manuscript and its Supporting Information files.

## Abstract

The hallmark of severe COVID-19 involves systemic cytokine storm and multi-organ injury including testicular inflammation, reduced testosterone, and germ cell depletion. The ACE2 receptor is also expressed in the resident testicular cells, however, SARS-CoV-2 infection and mechanisms of testicular injury are not fully understood. The testicular injury could be initiated by direct virus infection or exposure to systemic inflammatory mediators or viral antigens. We characterized SARS-CoV-2 infection in different human testicular 2D and 3D culture systems including primary Sertoli cells, Leydig cells, mixed seminiferous tubule cells (STC), and 3D human testicular organoids (HTO). Data shows that SARS-CoV-2 does not productively infect any testicular cell type. However, exposure of STC and HTO to inflammatory supernatant from infected airway epithelial cells and COVID-19 plasma decreased cell viability and resulted in the death of undifferentiated spermatogonia. Further, exposure to only SARS-CoV-2 Envelope protein caused inflammatory response and cytopathic effects dependent on TLR2, while Spike 1 or Nucleocapsid proteins did not. A similar trend was observed in the K18-hACE2 transgenic mice which demonstrated a disrupted tissue architecture with no evidence of virus replication in the testis that correlated with peak lung inflammation. Virus antigens including Spike 1 and Envelope proteins were also detected in the serum during the acute stage of the disease. Collectively, these data strongly suggest that testicular injury associated with SARS-CoV-2 infection is likely an indirect effect of exposure to systemic inflammation and/or SARS-CoV-2 antigens. Data also provide novel

**Funding:** This work was partially supported by grants NIH R21AI129465 (SV), NIH R21AI140248 (SV and HSA), Victoria S. And Bradley L. Geist Foundation (MDT and SV), George F. Straub Trust & Robert C. Perry Fund (MDT and SV), and NIH R01HD072380 and R03HD106936 (MAW). The funders had no role in study design, data collection and analysis, decision to publish, or preparation of the manuscript.

**Competing interests:** The authors have declared that no competing interests exist.

insights into the mechanism of testicular injury and could explain the clinical manifestation of testicular symptoms associated with severe COVID-19.

## Author summary

Testicular inflammation and germ cell depletion leading to decreased spermatogenesis is one of the complications of COVID-19 infection. Identifying testicular injury mechanisms is key to understanding the sequelae of SARS-CoV-2 on male reproductive health. We report that while human testicular cells do not support active SARS-CoV-2 replication, inflammatory media from infected airway epithelial cells and plasma from COVID-19 patients trigger cell death in multicellular 2D and 3D testis culture systems. Further, exposure of testicular cells to SARS-CoV-2 Envelope protein induces TLR2-dependent inflammatory cytokines and cytotoxicity. We also validate this finding in the K18-hACE2 mice that show testicular damage in the absence of replicating SARS-CoV-2. Our results identify both SARS-CoV-2 envelope protein and systemic cytokines as mediators of bystander testicular damage.

## Introduction

SARS coronavirus 2 (SARS-CoV-2), a positive-sense RNA virus, emerged in China in December 2019 and has since evolved into different variants and spread across the globe causing mild to severe coronavirus disease known as COVID-19. SARS-CoV-2 infects susceptible human cells by binding to Angiotensin-Converting Enzyme 2 (ACE2) and causes a range of clinical symptoms, which can progress to severe COVID-19 based on vaccination status and co-morbidities [1,2]. In addition to the acute lung injury with diffuse alveolar damage, other hallmarks of severe COVID-19 include multi-organ injury including vascular inflammation, cardiac complications, and kidney failure [3,4]. Epidemiological studies suggest that males irrespective of age and co-morbid conditions are disproportionately more strongly affected and present a higher case-to-fatality ratio than females [5].

Recent clinical findings report orchitis (inflammation of the testis associated with pain and discomfort) as one of the symptoms in almost a quarter of infected men [6,7]. Further, post-mortem analysis of testis from COVID-19 patients display signs of mild to severe testicular pathology, including testicular swelling, tubular injury, germ cell and Leydig cell (LC) depletion, and leukocyte infiltration in the interstitium [5,8,9]. In addition, alterations in male fertility parameters like reduced sperm count and testosterone levels, and dysregulated ratio of testosterone to luteinizing hormone (T/LH) have been reported in COVID-19 patients [10,11]. Further, erectile dysfunction is identified as one of the post-COVID-19 symptoms and viral particles have been documented in penile tissue even after 7 months after infection [7]. However, while the virus has not been detected in the semen in several studies [12,13], one study reported the presence of low levels of SARS-CoV-2 RNA in the semen of 4/15 of patients at the acute stage of infection [14]. Although the presence of viral RNA in the semen is not direct evidence of productive infection of SARS-CoV-2 in the testis, and the ability of the virus to gain access to the testis may be a rare event, collective data suggest that testicular injury is one of the complications of COVID-19.

SARS-CoV-2 viral replication and pathogenesis in extra-pulmonary organs are currently not well understood. Several studies demonstrate the presence of low-level viral RNA and

virus-like particles (VLPs) in many organs like the heart, kidney, testis, intestine, and brain [9,15–17]. SARS-CoV-2 VLPs, comprised of all major structural proteins including Spike (S), Nucleocapsid (N), Membrane (M), and Envelope (E), are abundantly secreted by the infected cells and can enter cells just like SARS-CoV-2 infectious virions [18]. In addition, SARS-CoV-2 proteins like S, N, and open reading frame 8 (ORF8) have been detected in the plasma of infected individuals illustrating antigenemia as one of the hallmarks of SARS-CoV-2 infection [19,20]. Exposure of both S and N proteins induced pro-inflammatory cytokines including interleukin 6 (IL-6) and tumor necrosis factor-alpha (TNF-α) in human macrophages [21,22]. The E protein is shown to form cation channels in the lipid bilayer and trigger the hyperin- flammatory response in human macrophages and mice [23]. While specific mechanisms by which SARS-CoV-2 cause testicular injury are still being characterized, the cytokine storm is considered to be the main driving factor of multi-organ damage [24].

In humans, high level of constitutive expression of angiotensin converting enzyme 2 (ACE2) is reported in the testis that regulates testosterone production and interstitial fluid vol- ume via modulating conversion of Angiotensin II to Angiotensin I [25,26]. The single-cell RNA-sequencing datasets from human testes revealed high expression of ACE2 in undifferen- tiated spermatogonia including spermatogonia stem cells, LC, and Sertoli cells (SC) [27]. How- ever, although transmembrane serine protease 2 (TMPRSS2) is expressed in most of the cell types in the body, there are conflicting reports on its expression levels and co-expression with ACE2 in different testicular cells [28,29]. The presence of ACE2 receptor in multiple resident cells hypothetically makes the testes a potential target for SARS-CoV-2 infection or endocyto- sis of VLPs. Alternatively, systemic cytokine storm may induce bystander testicular inflamma- tion, thus explaining the orchitis symptom observed in COVID-19 patients. Therefore, the question remains whether the gonadal injury is the direct or indirect consequence of virus infection in the testes.

We have previously established human multicellular 2D and 3D testicular organoid models to study Zika virus (ZIKV) persistence and immunity [30,31]. Here, using these well-estab- lished multicellular testicular cell culture models, we show that SARS-CoV-2 can enter testicu- lar cells, but cannot establish productive virus infection. However, exposure of testicular cells with inflammatory media from SARS-CoV-2 infected human airway epithelial cells led to apo- ptotic death of undifferentiated spermatogonia. Further, only exposure to SARS-CoV-2 E pro- tein but not S1 and N protein induced a pro-inflammatory response that correlated with severe cytotoxicity. We also examined and validated the bystander effect of SARS-CoV-2 infec- tion on testicular injury using K18-hACE2 mice. Our data collectively provide the first evi- dence that the testicular injury is not due to direct infection of SARS-CoV-2 but more likely an indirect effect of exposure to systemic inflammation and/or SARS-CoV-2 antigens.

## Results

### Human testicular cells do not support productive SARS-CoV-2 infection

We, and others, have previously shown that human testicular cells like LC and SC express ACE2 [26]. Therefore, we first determined the infection kinetics of SARS-CoV-2 in primary human SC and LC, 2D culture of mixed seminiferous tubule cells (STC, comprised of SC, peri- tubular myoid cells, and undifferentiated spermatogonia), and 3D human testicular organoids (HTOs, comprised of LC, SC, peritubular myoid cells and undifferentiated spermatogonia). Low levels of viral RNA in the range of log 2–3 genome copies were detected in all cell models, but the virus copies did not increase between 24 and 96 hrs post-infection (hpi, Fig 1A). Virus titers measured in the supernatant by plaque assay also did not show the presence of infectious virions in any cell types at any time point (Fig 1B). We further tested infection of SC and LC at

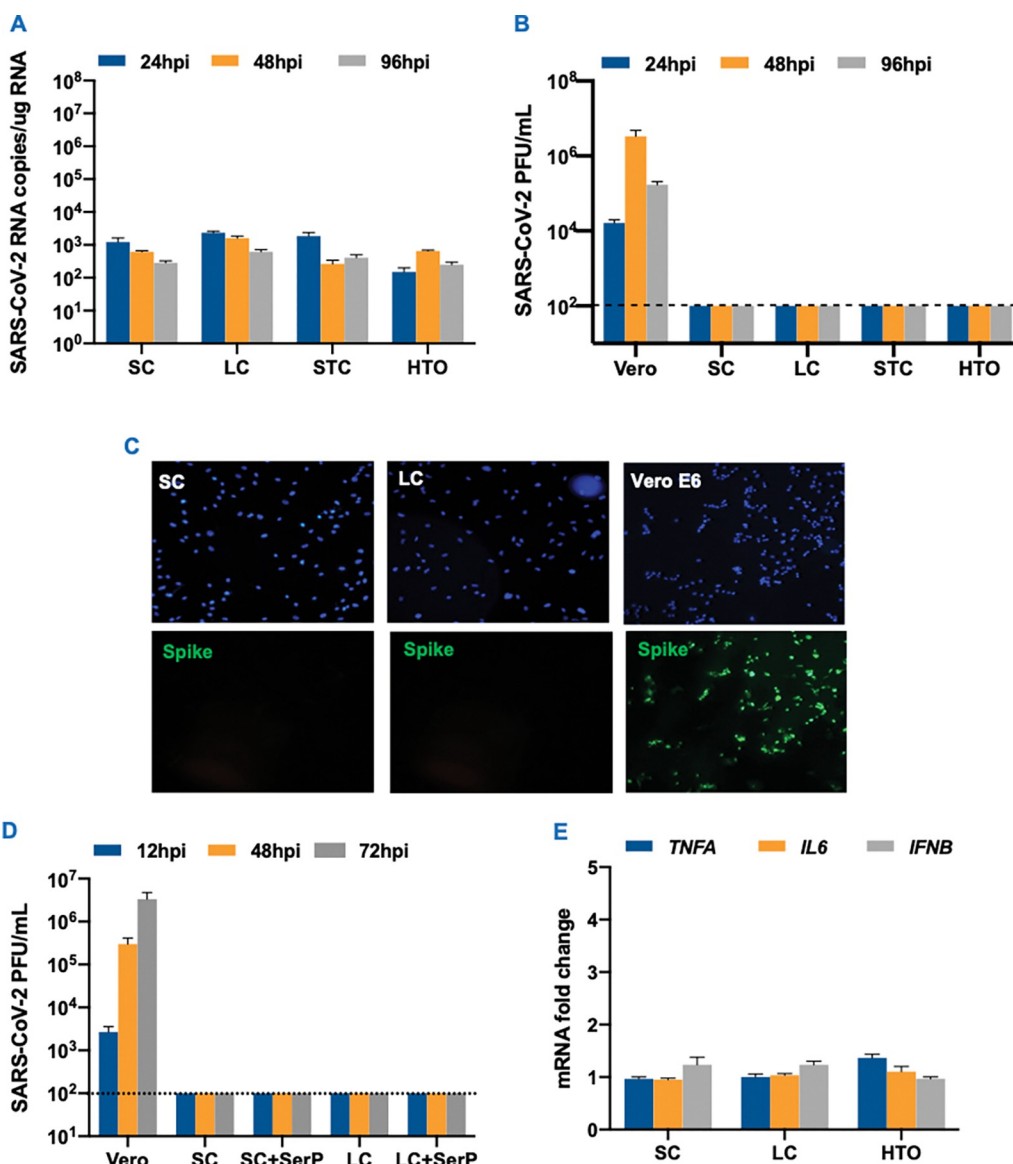

**Fig 1. Lack of SARS-CoV-2 does not establish a productive infection in human testicular cells.** (**A**) Primary SC, LC, STC and HTO were infected with SARS-CoV-2 at MOI 1 and intracellular virus levels were determined at 24, 48 and 96 hrs post-infection (hpi) using qRT-PCR. (**B**) SARS-CoV-2 titers in the supernatant from infected Vero E6, SC, LC, STC and HTO at MOI 1 measured using plaque assay. (**C**) Representative images of SARS-CoV-2 (MOI 1) infected SC, LC, and Vero E6 cells stained for SARS-CoV-2 using anti-Spike (green) at 48 hpi. (**D**) SARS-CoV-2 progeny titers measured by plaque assay in the supernatant from infected SC and LC pre-incubated with 5μg/mL of exogenous serine protease (SerP). (**E**) The mRNA fold change of *TNFA*, *IL6*, and *IFNB1* was measured in infected SC, LC, and HTO (MOI 1) using qRT-PCR at 48 hpi. The error bars represent the ±SEM of at least 4 independent infections. The dotted line represents the limit of detection (LOD).

higher MOI 10 that also did not show any SARS-CoV-2 released in the supernatant over 96 hours (S1A and S1B Fig). In addition, we were not able to detect the SARS-CoV-2 spike protein using immunofluorescence assay in infected SC and LC (Fig 1C), further demonstrating the lack of SARS-CoV-2 replication in these primary cell cultures.

Since exogenous serine proteases have been shown to facilitate SARS-CoV-2 entry and replication in other low TRMPSS2 expressing cells [32], we next assessed if TRMPSS2 is expressed

in LC and SC and if the presence of exogenous serine protease activity would make testicular cells susceptible to infection. We observed that there was minimal to no TMPRSS2 staining in these cells (S1C Fig). Assuming that this could be a factor explaining non-productive SARS-CoV-2 infection in these cells, we analyzed virus replication in the presence of exogenous serine protease. SARS-CoV-2 infection of SC pre-incubated with 5 μg/mL trypsin, a serine protease, which at this concentration does not interfere with cell attachment and has been used by others to enhance SARS-CoV-2 entry in other cell types [32], also did not result in increased infectious virions in the supernatant even at an earlier time point of 12 hpi (Fig 1D). To further evaluate if SARS-CoV-2 entry alone can activate an inflammatory response, we measured the mRNA levels of key cytokines associated with COVID-19 in SC, LC, and HTO at 48 hpi. Consistently, gene expression of inflammatory cytokines including *IL6*, *TNFA*, and interferon beta (*IFNB1)* was not altered in any of these testicular cell types (Fig 1E). Collectively, our data strongly suggest that even though SARS-CoV-2 RNA is detected in testicular tissue from COVID-19 patients, it does not establish a productive infection in resident human testicular cells.

## SARS-CoV-2 infection of human airway epithelial cells is associated with loss of air-liquid barrier integrity and production of inflammatory cytokines

Several clinical studies have linked SARS-CoV-2-associated cytokine storm with injury to the kidney, heart, and brain [24,33–36]. Therefore, we next tested if the testicular damage was an indirect effect of the inflammatory mediators derived from SARS-CoV-2 infection of other cell types. To begin the evaluation of the bystander effect, we first infected well-differentiated 2D cultures of human airway epithelial cells (HAE) grown on transwell inserts at MOI 1 and measured infectious virions released both on the apical and basal sides of the inserts. The plaque assay demonstrated a significant increase in the virus titers at 2 days post-infection (dpi) that peaked at day 3 and subsequently declined by >2 logs by 8 dpi (Fig 2A). A similar trend was observed in intracellular virus genome copies, with peak virus replication at 4 dpi (S2A Fig). The transepithelial electrical resistance (TEER) readings showed a decline starting at 3 dpi with significantly lower values at 5 and 6 dpi (S2B Fig), suggesting a loss in the barrier integrity most likely a result of virus-induced CPE. Peak virus titers also correlated with significant induction of key inflammatory cytokine genes like *TNFA* and *IL6*, and antiviral genes including interferon-induced protein with tetratricopeptide repeats one (*IFIT1*) at 4 dpi (Fig 2B). We further determined the secretion of multiple cytokines in both apical and basal media using multi-plex Luminex assay. As shown in Fig 2C, inflammatory mediators primarily associated with the cytokine storm in COVID-19 patients such as TNF-α, IL-1β, IL-6, IL-8, VEGF, and GM-SCF were elevated in both apical and basal media. The production of key cytokines in the HAE basal media was further compared to the plasma from COVID-19 patients during the acute stage of the disease (4–6 days of symptom onset) using ELISA. Interestingly, the levels of these cytokines at 4 dpi were comparable to the levels seen in the plasma of COVID-19 patients (Fig 3A–3C). Collectively, this data shows that basal media from SARS-CoV-2 infected HAE mimics the profile of select cytokines observed in COVID-19 patients and can be used to evaluate the indirect effect of infection on 2D and 3D human testicular models after UV-inactivation of the virus.

## SARS-CoV-2 infection-derived inflammatory mediators cause indirect toxicity on primary human testicular cells

To determine the cytotoxic effects of SARS-CoV-2 infection-derived inflammatory mediators on testicular cells, we exposed STC to UV-inactivated basal media from HAE and COVID-19

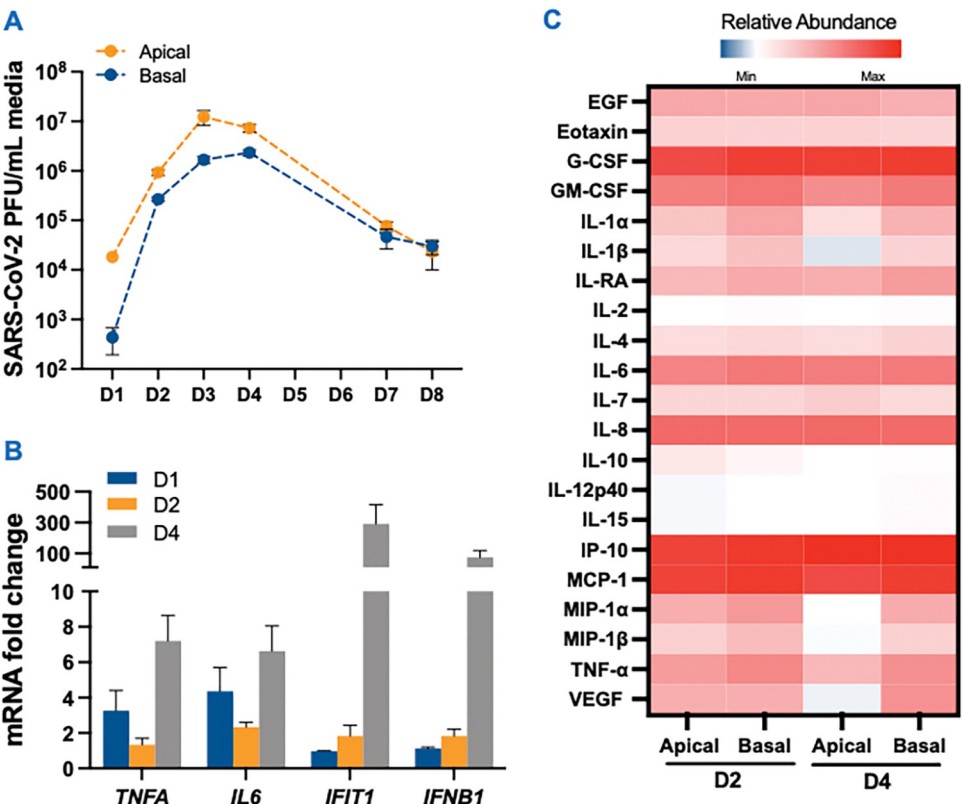

**Fig 2. Human airway epithelial cells are highly permissive to SARS-CoV-2 infection and produce inflammatory cytokines. (A)** Fully differentiated primary HAE grown on inserts were infected with SARS-CoV-2 (MOI 1) and infectious virions released on the apical and basal side at days (D) 1, 2, 4, and 8 post-infection were quantified using plaque assay **(B)** The mRNA fold-change of *TNFA*, *IL6*, *IFIT1* and *IFNB1* genes was measured in infected HAE at MOI 1 using qRT-PCR. **(C)** Heatmap depicting the levels of pro-inflammatory cytokines in the apical and basal media of HAE at indicated time points post infection. Data were log10 transformed and normalized to respective mock apical and basal media. Error bars represent ±SEM of at least 3 independent infections.

plasma after confirming the absence of virus RNA (S3A Fig) and measured the cell viability. As seen in Fig 3D, at 24hrs post-exposure, the cell viability of the STC exposed to infected HAE supernatant declined by approximately 30%. In contrast, the viability of cells exposed to supernatant from mock-infected HAE cells was comparable to untreated cells. Similarly, an almost 50% decrease in the viability of STC and HTO was observed when exposed to COVID-19 plasma as compared to healthy control plasma (Fig 3E). On the other hand, exposure of STC to conditioned media from SC infected with ZIKV, which establishes productive infection in multiple testicular cells, led to only 15% decline in cell viability (S3B Fig). To further understand if cell death following exposure to inflammatory plasma also triggers cytotoxic cytokines, we measured mRNA levels of *IL6*, *IL1B*, and *TNFA* genes. There was a significant increase in the transcripts of these cytokines as well as Bcl-2 associated protein X (*BAX)*, a pro-apoptotic gene, in HTO exposed to COVID-19 plasma (Fig 3G). Interestingly, however, SC alone exposed to infected HAE supernatant did not exhibit any significant change in cell viability at 24 hours post-exposure (Fig 3F). We also did not observe a similar induction of cytokines and *BAX* in SC following exposure to HAE supernatant (Fig 3H), suggesting that the source of cell death seen in our mixed 2D and 3D cultures was likely delicate germ cells.

To further validate that undifferentiated spermatogonia are more susceptible to cell death, we conducted a TUNEL assay on STC exposed to both UV-inactivated HAE supernatant and

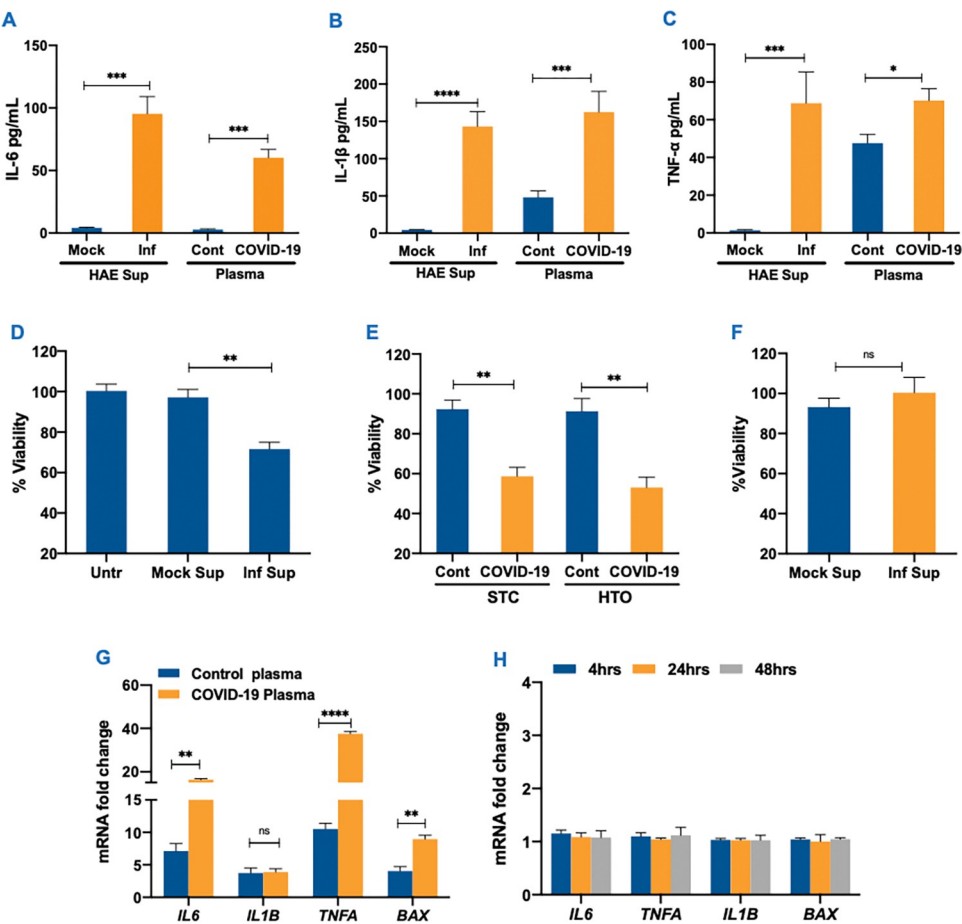

**Fig 3. Bystander effect of SARS-CoV-2 on different testicular cells. (A-C)** IL-6, IL-1β and TNF-α levels were measured by ELISA in the mock and UV-inactivated basal media from infected HAE cells (Inf) at 4 dpi, and control (healthy donor) and COVID-19 plasma **(D)** Percent cell viability assessed in STC at 24 hrs following exposure to UV-inactivated HAE infected (Inf Sup) or control supernatant (Mock Sup) **(E)** Percent viability of STC and HTO exposed to control (Cont) or COVID-19 plasma for 24 hrs post-exposure calculated by comparing to corresponding untreated cells (Untr) **(F)** Viability of SC exposed to UV inactivated HAE infected (Inf Sup) or control (Mock Sup) basal supernatant at 24hrs post exposure **(G)** The fold-change of *IL6*, *IL1B*, *TNFA*, and *BAX* transcripts in HTO exposed to control and COVID-19 plasma was measured using qRT-PCR (error bar represents ±SEM of 4 data points and each data point is a pool of RNA from 10 HTO) **(H)** The effect of exposure of HAE supernatant on the mRNA expression of *IL6*, *IL1B*, *TNFA* and *BAX* in SC was determined by RT-PCR. Error bars represent ±SEM of at least 3 independent exposures. *p<0.05; **p<0.01; ***p<0.001; ****p<0.0001.

COVID-19 plasma for 24 hours. As seen in Fig 4A, very few TUNEL-positive cells were detected in STC that were untreated or treated with mock HAE supernatant. However, TUNEL positive cells increased significantly (p<0.0001) to 10% in STC treated with infected supernatant for 24 hrs (Fig 4B). Similarly, TUNEL positive cells increased from 6% in healthy control plasma-treated cells to 20% (p<0.001) in COVID-19 plasma-treated STC (Fig 4A and 4B). The cells were also co-stained for UCHL1, a well-established undifferentiated spermatogonia marker, and merged images in Fig 4C show that UCHL1 positive cells were also TUNEL positive (yellow) in STC exposed to infected basal media from HAE cells. However, in STC exposed to COVID-19 plasma, we observed apoptotic cell death in both UCHL1 positive and UCHL1 negative cells (white arrows). These findings suggest that mediators derived from SARS-CoV-2 infection in the HAE media and COVID-19 plasma can cause an inflammatory response in testicular cells leading to apoptotic cell death.

## SARS-CoV-2 envelope protein causes severe damage in testicular cells

Virus-induced bystander cell death can be due to both inflammatory cytokines and/or viral proteins secreted by infected cells in the bloodstream, as shown in other viruses like Dengue and Ebola [37,38]. As a result, we investigated whether SARS-CoV-2 S1, N, and E proteins can cause cytopathic effects in various testicular cells. The SC were exposed to recombinant SARS-CoV-2 E, N, and S1 at different concentrations (0.25, 0.5, 1 and 4ng/μL media), and cell viability was quantified at 24hrs post-exposure. While the S1 and N proteins did not affect the cell viability, we observed a significant reduction in viability after exposure to the E protein in a dose-dependent manner with the most severe cell death seen in cells treated with 4 ng of E (Fig 5A). Further, to examine whether the cytotoxicity was specific to envelope protein, we pre-incubated envelope protein with proteinase K. SC death was reversed when E protein was inactivated with proteinase K indicating that E alone can cause SC death (Fig 5A). We then treated STC with E and S1 at concentrations 1 and 4 ng and measured cell death at 4 and 24hrs post-exposure and demonstrated a 30–40% decrease in cell viability only in E-treated STCs (Fig 5B). Similarly, E protein treated HTO showed a 40% reduction in cell viability at 24hrs post-exposure (Fig 5C). We also observed that exposure of STC to a high concentration of UV-inactivated virions ($10^7$ PFU/mL) did not induce any cytopathic effects (S4A Fig) suggesting that this response might be due to the free circulating proteins and not part of whole virus.

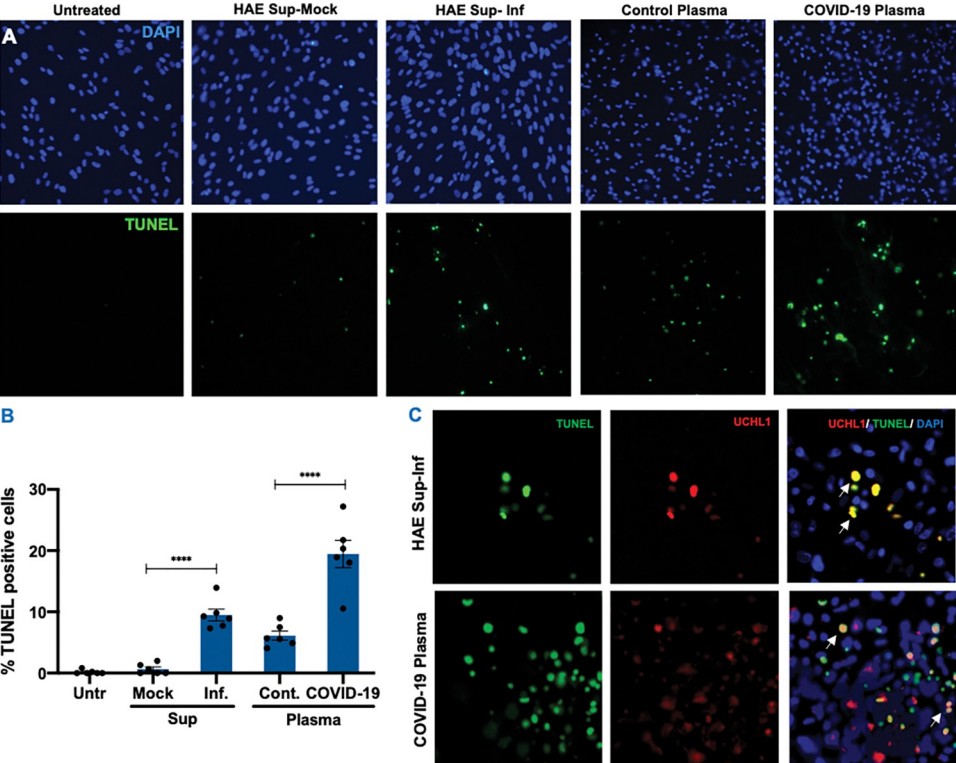

**Fig 4. SARS-CoV-2 infection-derived factors promote apoptotic cell death of undifferentiated spermatogonia. (A)** Representative TUNEL staining in STC exposed to supernatant from mock (HAE Sup-Mock) and infected HAE cells (HAE Sup-Inf), and to control and COVID-19 plasma for 24 hrs. The green fluorescence depicts TUNEL+ cells **(B)** Quantification of percent TUNEL positive cells and mean fluorescence intensity in each group. Data represents the average of at least six fields per coverslip from 3 independent experiments captured using Image J. Error bars represent ±SEM **(C)** STC exposed to UV-inactivated HAE supernatant and COVID-19 plasma were co-stained with TUNEL and UCHL1, a marker for undifferentiated spermatogonia. Co-localization was evaluated by merging TUNEL and UCHL1 and white arrows indicate overlapping green and red staining (yellow). ***p<0.001; ****p<0.0001.

To further validate if the cell death associated with the E protein is linked to cytokine induction, we measured the levels of cytotoxic cytokines like IL-6, TNF-α, and IL-1β in the supernatant of STC exposed to E and S1 proteins using ELISA. As seen in Fig 5D–5F, these cytokines were either absent or detected at very low levels in mock and S1-treated STC. However, their levels were significantly increased by E protein as early as 4hrs post-exposure and further increased at 24hrs post-exposure. The induction of IL-6, TNF-α, and IL-1β was also validated at the transcript level (Fig 5G), and the fold-increase of these cytokines at 24 hrs post-exposure to E protein in STC correlated well with the ELISA data. Moreover, we also observed a > 20-fold increase in the transcripts of the pro-apoptotic gene BAX (Fig 5G) that supported the cell viability data shown in Fig 5B.

Secretory virus proteins can activate inflammatory pathways following binding to cell surface receptors including TLRs [39]. Therefore, to examine the involvement of TLRs in the cytopathic effects associated with E, we exposed STC to E protein in the presence or absence of neutralizing antibodies against TLR1, TLR2, TLR4, and TLR6. As seen in Fig 5H, while the presence of antibodies against other TLRs did not affect the cell viability outcome, there was a nearly complete reversal in the cell death caused by E protein in the presence of anti-TLR2 (Fig 5H) suggesting that the downstream response of the E protein involves activation of the TLR2 receptor signaling. Furthermore, we tested if blocking TLR2 also reverses the cytokine response and found that *IL6. TNFA*, *IL1B*, and *BAX* transcript levels were significantly reduced in E protein treated SCs in the presence of anti-TLR2 (Fig 5I). Taken together, the data suggest that TLR2 senses SARS-CoV-2 envelope protein and plays a key role in initiating downstream pathogenic response.

To further determine if the cellular uptake of E protein is required for the cytopathic effects, we visualized STC exposed to E and S1 proteins conjugated with fluorophore AF488 using confocal microscopy. As seen in Fig 6A and 6B, E protein was internalized efficiently by STC after 12hrs of exposure and was detected mainly in the cytoplasm of these cells (white arrows). The intensity of the staining for Phalloidin, a marker for actin filaments, was significantly reduced (p<0.01) in E protein-treated STC (Fig 6C and 6D) indicating that the cytoskeleton was degraded. Further, the DNA material (blue arrows) was ejected from the nuclei of E protein-positive cells (Fig 6A), suggesting significant disruption of STC homeostasis. On the other hand, it appeared that the internalization of the S1 protein was significantly lower (p<0.05) than E protein and did not lead to disruption of the cytoskeleton or overall morphology (Fig 6C and 6D). This was in contrast with human macrophages where S1 was also internalized though not as efficiently as E protein but without a significant effect on the cytoskeleton (S4B Fig). Collectively, these data firmly establish that the SARS-CoV-2 E protein can cause severe cell death in the testicular cells that is dependent on TLR2 and is accompanied by an increase in the production of inflammatory cytokines.

## SARS CoV-2 infection of K18-hACE2 mice leads to testicular inflammation and injury

To characterize the effect of SARS-CoV-2 infection on the testis in vivo, we utilized the transgenic K18-hACE2 mouse model that expresses high levels of human ACE2 in the lung, low levels in the brain, and none in other epithelial cells like GI and liver [40]. Compared to other animal models like hamsters, ferrets, and primates, the K18-hACE2 mouse model best mimics different aspects of COVID-19 including severe lung disease, systemic cytokine storm, and tissue injury [41,42]. Since testicular injury is most commonly seen in moderate to severe COVID-19 patients, these mice are best suited for studying the indirect effects of SARS-CoV-2 infection on the testis. Intranasal inoculation of K18-hACE2 mice with $2x10^4$ PFU of

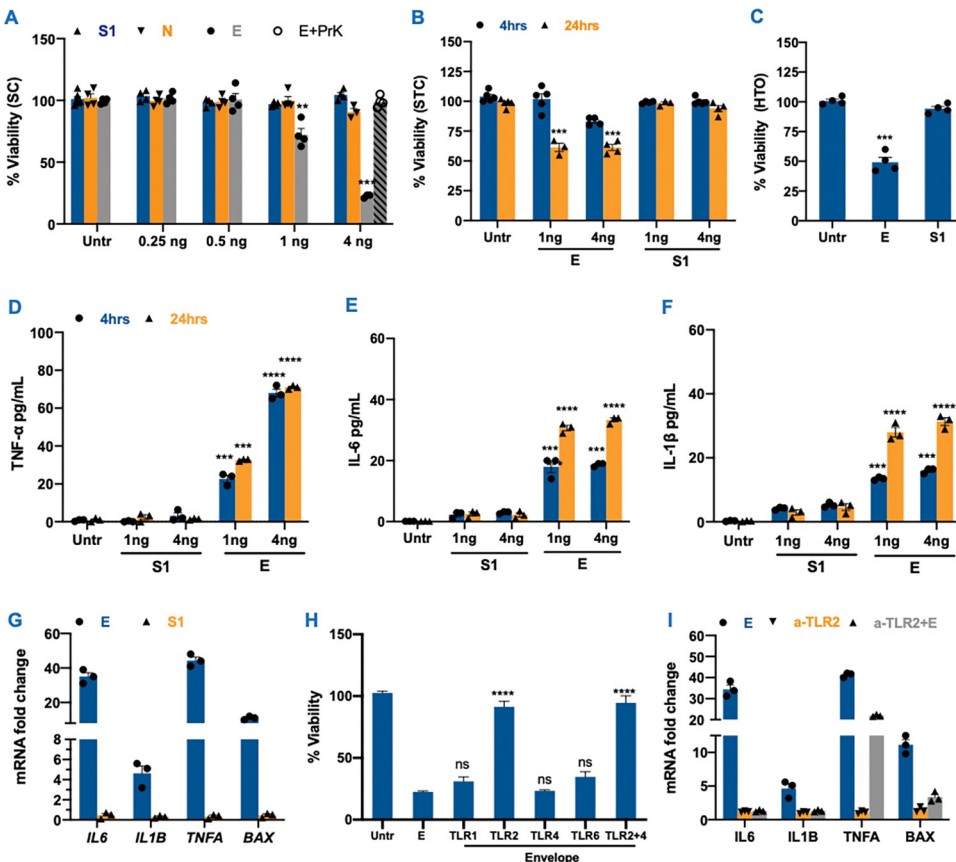

**Fig 5. SARS-CoV-2 Envelope protein triggers cell death and inflammation via TLR2. (A)** SC cell viability was assessed 24 hrs after exposure to recombinant SARS-CoV-2 spike subunit 1 (S1), nucleocapsid (N) and envelope (E) proteins at 0.25, 0.5, 1 and 4ng/µL media in 96-well plates. Exposure of E protein was also conducted in the presence of proteinase K enzyme **(B)** STC were exposed to S1 and E protein at 1 and 4ng/µL media and the percent change in cell viability was calculated after 24 hrs **(C)** Percent change in the cell viability of HTO was evaluated 24 hrs after exposure to recombinant SARS-CoV-2 E and S1 at 4ng **(D-F)** TNF-α, IL-6 and IL-1β levels in the supernatant from STC were measured using ELISA after exposure to recombinant E and S1 proteins **(G)** mRNA fold change of *IL6*, *IL1B*, *TNFA* and *BAX* in STC exposed to 4ng/µL of E and S1 proteins was measured at 24 hrs post-exposure using qRT-PCR. **(H)** SC were treated with E protein (4ng) in the presence or absence of neutralizing antibodies against different TLRs and percent cell viability was measured after 24 hrs of exposure. Error bars represent an average of at least 3–5 independent exposures **(I)** The fold-change in mRNA transcripts of *IL6*, *IL1B*, *TNFA*, and *BAX* was determined by RT-PCR in SC exposed to 4ng/µL of E protein in the presence or absence of anti-TLR2 antibody. Error bars represent ±SEM of at least 3 independent exposures. **p<0.01; ***p<0.001; ****p<0.0001 compared to respective untreated cells.

SARS-CoV-2 led to almost 70–80% mortality (Fig 7A) that replicated outcomes of other studies [42,43]. Virus genome copies in the lungs peaked at 3 dpi and remained high at 5 dpi (Fig 7B) and were almost cleared by 8 dpi. High virus titers were also observed in the brain which supports previous findings of neuroinvasion and encephalitis as a feature of this mice model [43,44]. However, virus mRNA was either found at very low levels in the heart in 50% of mice (Fig 7B). Interestingly, we also did not detect virus RNA in any of the testes at any time point (Fig 7B). Further, plaque assay of the tissue lysates demonstrated high virus titers present only in the lung and brain lysates at 3 and 5 dpi but not in heart and testis lysates (Fig 7C). We next assessed if inflammation markers are induced by the virus in the testes independent of active virus replication. As expected *IL6* and *TNFA* were significantly upregulated in the lungs and correlated with viral titers. Interestingly, despite no viral RNA, we found that the transcripts of these inflammatory cytokines increased significantly in the testes at 5 and 8 dpi

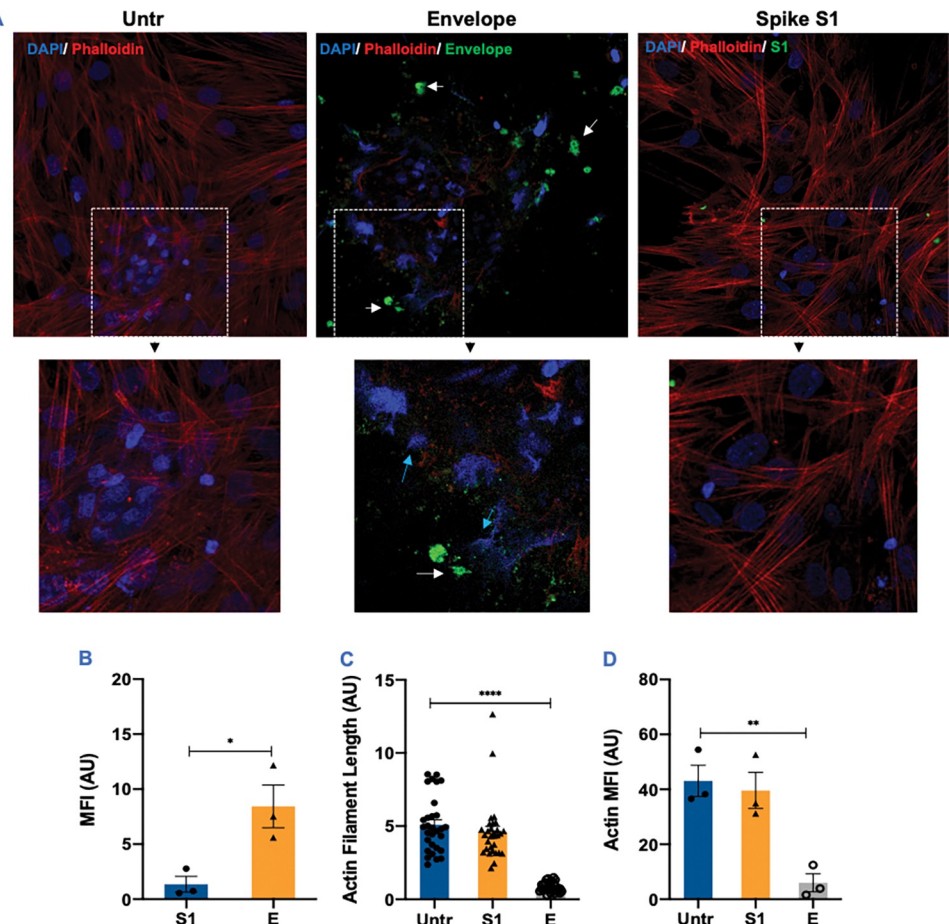

**Fig 6. The uptake of SARS-CoV-2 envelope protein disrupts the morphology of the seminiferous tubule cells.** STC were exposed to green fluorophore-conjugated E and S1 proteins (4ng) and the uptake was evaluated after 24 hrs by detecting intracellular virus antigens (green) following staining with DAPI and Phalloidin (red), a marker for actin filaments. **(A)** Representative confocal microscopy image show E protein localization in the cytoplasm (white arrows) and dramatic loss of actin filaments. High-power magnification pictures depict dramatic disruption of the nuclear compartment with genetic material being ejected from the nucleus (blue arrows). **(B)** The mean fluorescence intensity (MFI) of intracellular, **(C)** actin filament length and **(D)** MFI were assessed in untreated (Untr), S1, and E proteins exposed cells using Image J in 3 different fields from at least 3 independent experiments. AU = arbitrary units. *p<0.05; **p<0.01; ****p<0.0001.

(Fig 7D). To understand if peak virus infection in the lung also leads to the release of virus antigens in the serum as shown in COVID-19 patients, we measured S1, N and E levels during the asymptomatic phase of infection (3 dpi), acute stage of disease (5 dpi) and after virus clearance in survivor mice (8 dpi). While N protein was detected only at 5 dpi, S1 was present in mice at all time points (Fig 7E) with a peak at 5 dpi (p<0.01). Interestingly, E protein was detected at low levels at 3 dpi but the levels were high in 2/5 mice at 5 dpi (Fig 7E) suggesting that the release of virus antigens is not an event associated with every infected mice or these proteins are present only transiently during infection.

To further assess if there is an indirect effect of SARS-CoV-2 infection on the testes, we conducted a histopathological assessment of PAS-H-stained sections of the testes. The uninfected control males had normal testis and tubular organization (Fig 7F, i-ii). Seminiferous tubules were well-developed and tightly packed, with limited interstitial space. Tubular basal and adluminal compartments were tightly connected, and germ cells were properly organized,

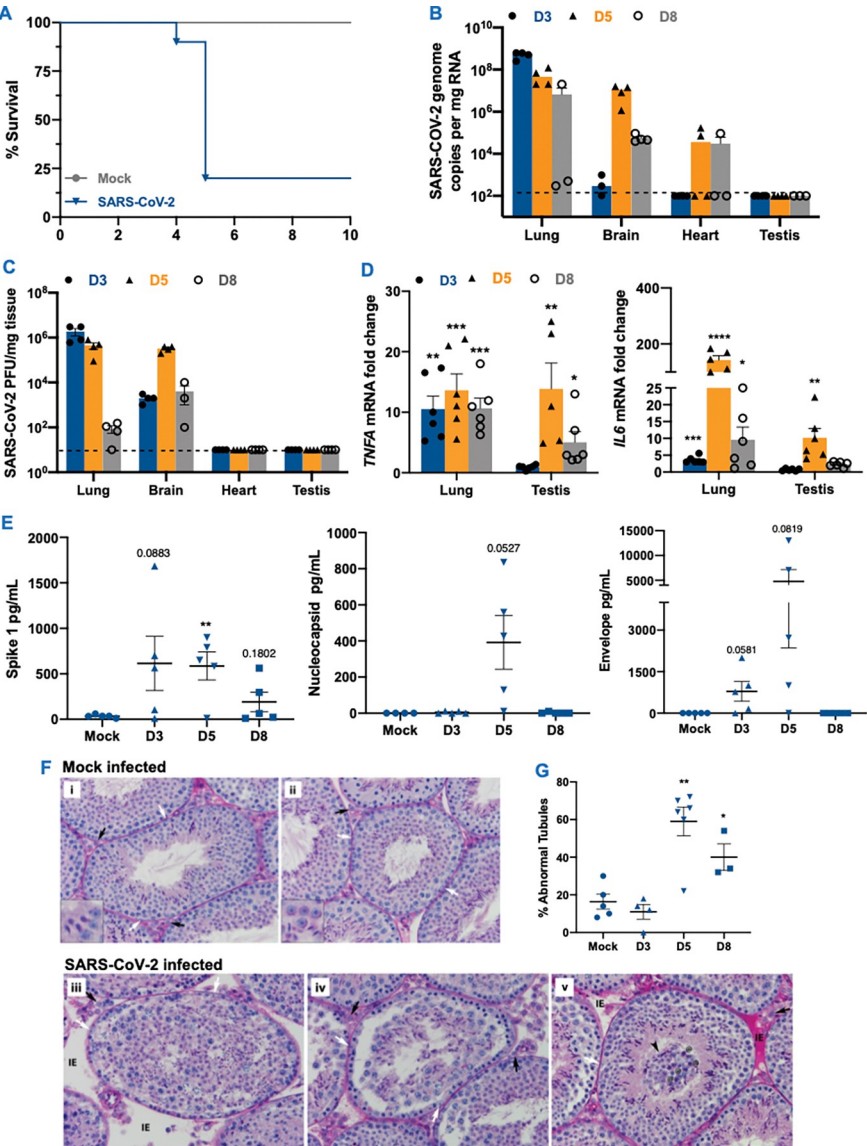

**Fig 7. SARS-CoV-2 infection in K18-hACE2 mice exhibits severe testicular pathology. (A)** Eight to twelve weeks old male and female K18-hACE2-transgenic mice were inoculated via the intranasal route with 104 PFU SARS-CoV-2. Survival was monitored for 14 days (n = 15) **(B)** Viral RNA in the lung, heart, and testis at days 3, 5, and 8 post-infection (dpi) measured by RT-qPCR. The dotted horizontal line indicates the limit of detection **(C)** Plaque assay was used to determine SARS-CoV-2 titers in the lung, heart, and testis tissue homogenates **(D)** Fold change in the gene expression of inflammatory genes *IL6* and *TNFA* was determined in the lung and testis homogenates using qRT-PCR (two independent experiments; n = 4–6 males per time point) **(E)** SARS-CoV-2 Spike (S1), Nucleocapsid (N) and Envelope (E) protein levels in the serum of K18-hACE2 mice were measured using ELISA **(F)** PAS-staining of testis sections from K18-hACE2 mice following mock (i-ii) or SARS-CoV-2 infection at 5 (iii, v) and 8 (iv) dpi. Seminiferous tubules from control mice show normal tubular morphology and healthy Leydig (black arrows) and Sertoli cells (white arrows). Insets show healthy round and elongated spermatids typical of the stage. In sections from SARS-CoV-2 infected mice various seminiferous epithelium abnormalities were observed including interstitial edema (IE in v), lack of lumen and overall cell disorganization (iii), separation of germ cell layers from the basal membrane (iv), sloughing of healthy and apoptotic germ cells into the lumen (v, arrowhead). Insets, 3x magnification. Scale, 100 μm. **(G)** Percent abnormal tubules based on the analysis of 50 randomly selected seminiferous tubules from each male. Each data point represents data from one mouse (n = 3–6 males per time point). *p<0.05; **p<0.01; ***p<0.001; ****p<0.0001 using unpaired t-test compared to respective mock-infected controls.

with spermatogonia, spermatocytes, and round spermatids visible at successively higher levels within the epithelium. Leydig cells in the interstitium and Sertoli cells and germ cells within tubules were normal and healthy. However, after SARS-CoV-2 infection, various testicular abnormalities were noted in both the interstitium and seminiferous tubules at 5 and 8 dpi. Interstitial edema of varying severity levels was observed in some areas, either as increased interstitial space between adjacent seminiferous tubules (Fig 7F, iii at 5 dpi) or with red-stained fluid filling the interstitial space (Fig 7F, v at 5 dpi). Leydig and Sertoli cells, as well as most germ cells, appeared healthy. However, in some areas within seminiferous tubules, germ cells were severely disorganized, in extreme cases, randomly occupied space throughout the tubule (Fig 7F, iii, 5 dpi). In some areas, the tubules were found to be congested with evidence of prematurely sloughed germ cells in the lumen (Fig 7F, v at 5 dpi). The quantitation of abnormal tubules based on the features of cells in the lumen, lack of clear lumen, separation from the basement membrane, and apoptotic cells, as shown previously [45], indicated that while the percent tubular defect at 3dpi was comparable to controls, it increased significantly by 3–4 folds in testis sections at 5 and 8 dpi (Fig 7G). These results validate our in vitro data and collectively demonstrate that despite no active replication, SARS-CoV-2 infection results in interstitial and tubular abnormalities in the testis of hACE2 mice. These injury markers are similar to what has been observed in humans [8,9,46], suggesting that these mice can be used as a model to systematically delineate the indirect effect of SARS-CoV-2 on different aspects of male reproductive health.

## Discussion

Emerging clinical studies highlight that SARS-CoV-2 infection-associated testicular pain, reduced testosterone levels and altered sperm counts are more common in COVID-19 patients than previously thought [5,8,9]. Further, postmortem studies have characterized several features of testicular injury including the detachment of SC, apoptosis of undifferentiated spermatogonia, and infiltration of leukocytes in the interstitium. However, the association of these injury markers with virus infection kinetics is not clear. Here, we used different 2D and 3D culture models of primary human testicular cells to show that (i) while SARS-CoV-2 can enter LC and different seminiferous tubular cells, it cannot establish a productive infection in any of these cell types, (ii) Inflammatory media from infected airway epithelial cells and plasma from COVID-19 patients can trigger inflammatory cytokines production and cytotoxicity in testicular cells, (iii) exposure of testicular cells to SARS-CoV-2 E protein increases expression of inflammatory cytokines and induce severe cytotoxicity that is dependent on TLR2, and (iv) intranasal inoculation of K18-hACE2 mice depicted leads to testicular damage in the absence of any replicating virus, thus overall supporting the fact that testicular damage is a bystander effect of SARS-CoV-2 infection.

Susceptibility to SARS-CoV-2 infection is highly cell type-specific and dependent on the presence of entry receptors like ACE2 and serine protease, TMPRSS [47]. While robust infection of lung alveolar type II epithelial cells is linked to high ACE2 expression, the absence of this receptor and serine proteases is the main reason for the human macrophages, natural killer (NK) cells, dendritic cells, and vascular endothelial cells not being susceptible to SARS-CoV-2 [27,48]. Enterocytes in the gastrointestinal (GI) tract that express very high levels of ACE2 and TMPRSS2 are susceptible to SARS-CoV-2, but the virions produced are very low compared to alveolar type II cells [49,50], suggesting that just the presence of receptors alone is not enough to establish productive viral infection. Although several groups, including ours, have reported high levels of ACE2 in the human testes including LC, SC, and undifferentiated spermatogonia [27,29], direct evidence of infection of human testicular cells is lacking. The

presence of SARS-CoV-2 RNA in the human postmortem testes tissue is not a common observation and is limited to RT-PCR detection of very low levels of viral genome copies [8]. Even in the animal models, subgenomic SARS-CoV-2 RNA was detected only in the intratesticular inoculated hamsters [51]. Further, the suggestion that SARS-CoV-2 can infect testes of the rhesus macaques by Madden and colleagues was based on the staining of Spike protein [52] and does not confirm if testis can support active replication of the virus. Therefore, taken together, our data showing the total absence of SARS-CoV-2 virions in the media and no cell death in infected cells, provide direct evidence that SARS-CoV-2 cannot establish productive replication in different testicular cells in vitro. Since virus replication is directly associated with robust inflammatory response, the absence of the induction of the key cytokines in infected cells again supports our notion that none of our *in vitro* testicular cell models supported virus replication. We speculate this can be either due to the absence of co-localization of TMPRSS2 and ACE2 in the same cell type or the lack of specific host components required for virus replication.

However, robust data now exists suggesting that COVID-19 leads to severe testicular injury and affects testis function [8]. In the absence of active virus replication, tissue injury can be mediated by cytokines storm or exposure to virus proteins [23]. Mild and severe COVID-19 patients exhibit systemic cytokine profiles similar to other infectious diseases such as Ebola virus disease (EVD) [53]. Systemic cytokine storm is also well described in influenza-infected patients and a recent study compared the cytokine profile with COVID-19 patients [54]. The data shows that cytokines like IL-6, TNF-a, and IL-1β are produced by both the influenza virus and SARS-CoV-2, though levels are higher in COVID-19 patients. Further, IL-4, IL-9, CCL5, CCL8, GM-CSF, and PDGF are exclusively induced by SARS-CoV-2 and these differences may explain why multi-organ injury including testicular damage and long-term sequelae is observed in COVID-19 patients and not with influenza virus. The elevated levels of TNF-α, IL-6, IL-1β and IFNγ reported in severe EVD are associated with severe damage to the kidney and vascular system [55]. Similarly, dengue nonstructural protein (NS1) shed during acute infection acts as a viral pathogen-associated molecular pattern that activates TLR4 on leukocytes and endothelial cells leading to inflammation and endothelial dysfunction [37]. Therefore, we next focused on addressing our alternative hypothesis that testicular injury results from the bystander effect of systemic cytokines. Our observation of the comparable levels of cytokines in HAE media and COVID-19 plasma is encouraging and supports the notion that inflammatory HAE basal media can be used to study bystander effects of SARS-CoV-2 associated cytokine storm. Virus replication, inflammatory response, and transmigration of SARS-CoV-2 from the apical to the basal side of the inserts have been reported previously, but our data here demonstrate the in-depth comparison of cytokines and chemokines released in both apical and basal media and also show a strong correlation between key inflammatory cytokines in the supernatant from HAE cells and COVID-19 plasma. Interestingly, both STC and HTOs exhibited significantly higher cytotoxicity post-exposure to inflammatory media from infected HAE and plasma from COVID-19 patients compared to SC (Fig 3). Our speculation that this difference is most likely because of the presence of delicate undifferentiated spermatogonia in the STC and HTO was confirmed by the TUNEL assay and agrees with germ cell depletion seen in the testis from COVID-19 patients [6].

While the secretion of E protein by infected cells in COVID-19 patients is not yet determined, recently the presence of SARS-CoV-2 S1 and N proteins in the plasma of ~60% of COVID-19 patients has been shown in the range of 5–10,000 pg/mL plasma [56]. Other studies also report the presence of S1 in the urine and saliva of COVID-19 patients [57] suggesting that the shedding of different SARS-CoV-2 proteins is an outcome of infection and might be an event associated with severe disease. Additionally, the presence of VLPs in different tissues

including the testes and the brain is also commonly reported [17]. Both S1 and N proteins have been shown to induce inflammation and cell death in macrophages [21,22], suggesting the potential of these proteins to independently cause cytopathic effects. On the other hand, Zheng and colleagues reported that exposure to S1 did not induce any inflammatory response compared to E in bone marrow-derived macrophages (BMDMs) [58]. Therefore, although it was surprising that S1 did not induce any cytopathy in SC and STC (Fig 6), we believe this might be because S1 was not efficiently internalized in these testicular cells compared to more phagocytic macrophages as shown in our data (S4B), which may partly explain the disparity seen in our and other studies. This may also indirectly support the notion that vaccine-associated Spike antigen production may not have a significant effect on the testis. Our data, however, agrees with previous in vitro and in vivo studies that also showed induction of cytokine response, cell death, and lung pathology by SARS-CoV-2 E protein and dependence on the TLR2 pathway [58,59]. The reversal in the transcript levels of cytokines in the presence of anti-TLR2 (Fig 5I) further suggests the involvement of E-TLR2 interaction in activating downstream inflammatory pathways including AKT or ERK signaling. Mouse testicular cells including Sertoli cells do express different TLRs and the role of TLR2 in mumps virus-associated inflammation is well established [60]. There is a consensus view that E, a glycosylated transmembrane protein with ion channel activity, plays an important role not only in viral replication and virion assembly but also in pathogenicity including induction of cytokines and cell death [23]. These studies and our data collectively suggest that E protein can trigger an inflammatory response and cause cytopathic effects in the testicular cells independent of SARS-CoV-2 replication. However, since E protein levels are not yet reported in the plasma of COVID-19 patients, the relative contribution of the E protein and inflammatory mediators in inducing cytotoxicity cannot be delineated at this point.

An important highlight of our study is the validation of the in vitro data in the K18-hACE2 mouse model. Since high expression of hACE2 is mainly in the lungs, as expected very high viral replication was detected in this organ leading to high mouse mortality. Elevated levels of inflammatory cytokines have been reported before not only in the lungs but also in the plasma of these mice [61]. Therefore, we believe that the K18-hACE2 mouse model is an appropriate model for studying the bystander effect of SARS-CoV-2 infection. Our data provide the first evidence that the testicular pathological events similar to what is reported in postmortem testis tissue from COVID-19 patients [8] manifest in the K18 hACE2 mice. Seminiferous tubule disorganization, germ cell sloughing, and germ cell apoptosis that we observed in mouse testis sections are well-characterized hallmarks of testicular injury thus establishing K18-hACE2 mice as a tool that would allow systematic delineation of underlying mechanisms at the molecular level in future studies. Our data also provide the first evidence of the levels of virus antigens in the serum of infected mice. The levels of S1 follow a similar trend seen in humans with a peak during the symptomatic phase that declines after recovery [56]. The detection of the E protein in the sera is an important observation as it suggests that this protein is secreted by infected cells although the concentration varied a lot between mice. However, since E protein levels are not yet reported in COVID-19 patients yet, we speculate that systemic inflammation is the main driver of testicular injury, and virus antigens may act as another contributory factor to this damage depending on their levels.

In summary, despite reported gross testicular pathology and alterations in male reproductive health including lower testosterone levels and decreased sperm count in COVID-19 patients, our understanding of the mechanisms of SARS-CoV-2-associated testis injury is limited. Few studies have reported the presence of viral antigens and/or VLPs in the testis by immunostaining, based on which it has been proposed that testicular injury is the result of direct SARS-CoV-2 infection [62]. However, our data present direct evidence that despite the

expression of ACE2, human testicular cells do not support productive infection of SARS-CoV-2. This study not only improves our understanding of the indirect effect of virus infection on the testis but also suggests a potential role of virus proteins in mediating pathology not only in the testis but also in other organs including the kidney, brain, and heart. During peak infection, exposure of testicular cells to both cytokine storm and viral antigens may trigger pathological pathways and apoptotic death of germ cells that may be responsible for orchitis symptoms and lower sperm counts reported in COVID-19 patients. Finally, our findings presented herein suggest the need for long-term follow-up of male reproductive health markers in moderate to severe COVID-19 cases following recovery including patients suffering from long-COVID.

## Materials and methods

### Ethics statement

Human adult testicular tissues from brain-dead patients were procured through the National Disease Research Interchange (NDRI) and used under the regulation and approval of the institutional review board of Wake Forest School of Medicine. Animal procedures were performed in accordance with the national, and institutional guidelines for the care and use of animals and approved by the Institutional Animal Care and Use Committee of the University of Hawaii at Manoa (Protocol# 21-3597-2).

### Cells, Testicular Organoids, and virus infection

Primary human SC and LC were obtained from iXCells Biotechnologies and ScienCell Research Laboratories, respectively. Low passage SC and LC were cultured in DMEM/F-12 and Leydig Cell Medium as described previously [31]. The human testicular organoids (HTO) consisting of primary SC, LC, peritubular myoid cells, and undifferentiated spermatogonia were generated from adult human testicular tissue procured through the NDRI and cultured in ultralow-attachment 96-well round-bottom plates as described by us previously [63]. For mixed seminiferous tubule cells (STC) culture, seminiferous tubules from adult testes were digested to isolate mixed cell populations of SC, peritubular myoid cells, and undifferentiated spermatogonia as described previously [30,64]. SARS-CoV-2 USA-WA1/2020 strain was obtained from BEI Resources, propagated once in Vero E6 cells, and used for all in vitro experiments. Cells cultured in 6-, 24-, or 96-well plates were infected with SARS-CoV-2 at MOI 1 or 10 and incubated for 1.5 hrs at 37% and 5% CO2. HTO were infected using $10^4$ PFU SARS-CoV-2. All SARS-CoV-2 manipulations were performed in the dedicated BSL3 facility at the John A. Burns School of Medicine.

### Virus quantitation

SARS-CoV-2 titers in cell culture supernatants were measured by plaque assay using Vero E6 cells and expressed as PFU per mL of supernatant [65]. Intracellular viral genome copies were measured in the RNA extracted from cell lysates and tissue homogenates at different time points post-infection by qRT-PCR. Forward (nCoV_IP4-14059Fw: GGTAACTGGTATGAT TTC G) and reverse (nCoV_IP4-14146Rv: CTGGTCAAGGTTAATATAGG) primers and probe (nCoV_IP4-14084Probe(+): TCATACAAACCACGCCAGG [5']Fam [3']BHQ-1) were used specific for SARS-CoV-2 RNA-dependent RNA polymerase gene region and expressed as genome copies per μg of RNA [66].

## Exposure of testicular cells to inflammatory media, COVID-19 plasma, and SARS-CoV-2 proteins exposure

Human airway epithelial cells (HAE) grown on the inserts were infected with SARS-CoV-2 at MOI 1 and the media was collected from the basal and the apical side of inserts at different time points after infection. The basal side supernatant was treated with ultraviolet light (UV) for 12 min to inactivate infectious virions. Different testicular models were exposed to UV-inactivated HAE basal media (1:1 ratio with cell culture media). SC and STC cultures were also exposed to SARS-CoV-2 E, N, or S1 proteins at 1 or 4 ng/μL concentration. These proteins, obtained from ProSci Incorporated, were generated in E.coli cells, purified by SDS page, and certified as endotoxin-free. Plasma from 5 RT-PCR+ COVID-19 patients collected during the symptomatic phase (days 4–6 of symptoms) under the UH IRB# 2020–00367 and age-matched healthy controls were used in this study. STC and HTOs were exposed to plasma (1:5 ratio with cell media that was determined by viability assays using serial dilution of plasma) and cell viability and TUNEL staining assays were conducted after 24 hrs of exposure. E protein was also incubated with 1X proteinase K for 3 hrs before exposing to SC. In some experiments, E and S1 exposure were conducted in the presence or absence of neutralizing antibodies against TLR1, TLR2, TLR4 and TLR6 (ThermoFisher Scientific Cat. # 14-9911-80 Cat. # MA5-16200, Cat # MA180122, and Cat. # MA516173 respectively at 1:250 or 1:500 concentration). In parallel, human macrophages grown on coverslips were exposed to S1 and E protein at 1 ng/μL and visualized for protein uptake by confocal microscopy.

## RT-PCR analysis

Total RNA was extracted from mock- and SARS-CoV-2-infected SC, LC, STC, and HTO lysates using RNeasy mini kit (Qiagen) and synthesized into cDNA, and change in mRNA transcripts of inflammatory genes was measured by qPCR, as described previously [67]. Specific primer sequences are either previously described [31,68] or shown in Table 1. The housekeeping gene *GAPDH* was used to normalize fold change values of antiviral genes, with respective controls used as a reference control.

## Cell Viability

Cell viability of different 2D cultures at different time points of infection or exposure to virus proteins or HAE supernatant was determined using the CellTiter 96 AQueous One Solution

**Table 1. Primer sequences used for qRT-PCR.**

| Gene Name (Human) | Direction | Sequence 5'-3' |
|---|---|---|
| ACE2 | F | GGACTGATATAGGAAGGATAG |
| | R | ACCTTTGAAGAGATTAAACC |
| BAX | F | CTGACGGCAACTTCAACTGG |
| | R | TCTTGGATCCAGCCCAACAG |
| Gene Name (Mouse) | Direction | Sequence 5'-3' |
| GAPDH | F | TTGTCTCCTGCGACTTCAAC |
| | R | GTCATACCAGGAAATGAGCTTG |
| IL6 | F | TAGTCCTTCCTACCCCAATTTCC |
| | R | TTGGTCCTTAGCCACTCCTTC |
| TNFA | F | CACCACGCTCTTCTGTCT |
| | R | GGCTACAGGCTTGTCACTC |

cell proliferation assay (G3582; Promega), while HTO viability was determined by the CellTiter-Glo 3D cell viability assay (Promega G9681; Promega) as described previously [31].

## Enzyme-Linked Immunosorbent and Luminex Assays

All commercially available ELISA kits were purchased from Invitrogen Thermo Fisher Scientific and the assays were performed according to the manufacturer's instructions. The kits used were IL-6 Human Instant ELISA Kit (Cat. # BMS213INST), IL-1β Human Instant ELISA Kit (Cat. # BMS224INST), and TNF-alpha human Instant ELISA Kit (Cat. # BMS223INST). All samples including HAE supernatant at different time points and plasma from COVID-19 patients were run in triplicate and levels were expressed as pg/mL media or plasma. Cytokines in the supernatant of HAE were measured using Millipore Human Cytokine/Chemokine MAGNETIC BEAD Premixed 29 Plex Kit (Cat. # HCYTMAG-60K-PX29) using Luminex 200 Miliplex Analyzer system. ELISA kits to measure S1, E, and N were purchased from Thermo Fisher Scientific (Cat # EH492RB), EagleBio (Cat # KBVH015-30), and BioLegend (Cat # 448007) respectively as assays were conducted as per the instructions.

## Immunofluorescence and TUNEL assay

Mock and infected LC, SC, or STC grown on glass coverslips were fixed with 4% PFA, permeabilized with 0.1% Triton X-100 in PBS, and blocked with 5% bovine serum albumin in PBS. Cells were then incubated with primary antibodies against anti-Spike (GeneTex, GTX632604 at 1:500 dilution), followed by fluorophore-conjugated secondary antibody (Invitrogen Alexa Fluor 488-conjugated sheep anti-rabbit, 1:5000 dilution), and examined using an Axiocam MR camera mounted on a Zeiss Axiovert 200 microscope. TUNEL assay was performed using the Promega Fluorometric TUNEL System according to the manufacturer's instructions. Undifferentiated spermatogonia were also stained for the well-established cell-specific marker, ubiquitin carboxyl-terminal esterase L1 (UCHL1) [31] using rabbit anti-human UCHL1 (Sigma, HPA005993 at 1:1,000 dilution). The secondary antibody was Alexa Fluor 594-conjugated goat anti-rabbit (Invitrogen; 1:5000 dilution).

## Infection of *K18-hACE2* mice

*B6.Cg-Tg (K18-ACE2)2Prlmn/J* (*K18-hACE2*) mice (strain #034860) were obtained from the Jackson Laboratory. All mouse infection experiments were conducted at the dedicated ABSL2/3 facility at the UH using SARS-CoV-2 USA-HI-B.1.429 isolated from a local COVID-19 patient in 2020 that is very similar to the SARS-CoV-2 CoV/USA-WA1/2020 [69] at the dedicated ABSL2/3 facility at the UH. Eight to twelve weeks old *K18-hACE2* mice were inoculated with $2x10^4$ PFU SARS-CoV-2 via the intranasal route and observed daily to record body weights and clinical symptoms and were sacrificed when weight loss greater or equal to 20% was observed. The lung, brain, heart, and testis tissues were harvested in a separate set of experiments at 3, 5, and 8 days post infection (dpi) and were either flash-frozen or fixed in 4% PFA to determine virus genome copies and histopathological changes respectively. RNA was extracted from frozen tissues as described previously [65] and virus RNA and expression of different host genes were measured by RT-PCR. Testes were also fixed in Bouin overnight, and then stored in 70% ethanol prior to embedding in paraffin wax, sectioning at 5 μm, and staining with Periodic acid Schiff and hematoxylin (PASH) to identify histopathological changes.

## Statistical analysis

All data were analyzed with GraphPad Prism 9.3.1 software. Statistically significant differences between different groups were determined using unpaired t-tests. SARS-CoV-2 titers and viability data are reported as means +/- standard error of the mean (SEM) from at least three or more independent experiments. Gene expression (mRNA fold change) and ELISA data are reported as means +/- SEM from ≥3 independent experiments. A *p-value* of <0.05 was considered statistically significant for all analyses.

## Supporting information

**S1 Fig.** Primary SC, LC, STC, and HTO were infected with SARS-CoV-2 at MOI 10 and **(A)** intracellular virus levels were determined using qRT-PCR and **(B)** virus titers in the supernatant were measured using plaque assay **(C)** Representative TMPRSS2 (green) and DAPI (blue) staining of primary Sertoli (top) and Leydig (bottom) cells.
(TIFF)

**S2 Fig.** **(A)** SARS-CoV-2 RNA was measured in HAE cells at days (D) 1, 2, 4 and 8 post-infection using qRT-PCR **(B)** The transepithelial electrical resistance (TEER) was used to measure the integrity of the air-liquid barrier of HAE inserts at different days post-infection (MOI 1), and expressed in Ohm*cm$^2$ ($\Omega$cm$^2$) **p<0.01; ****p<0.0001 compared to mock.
(TIFF)

**S3 Fig.** **(A)** Progeny virus titers in the basal media of HAE cells (2 dpi as control) and serum of COVID-19 patients demonstrating no virus in the patient sera **(B)** STC were exposed to UV-inactivated supernatant from dengue virus (DENV)- infected HUVEC and Zika virus (ZIKV) infected SC and viability was calculated after 24hrs. **(C)** SC were exposed to UV-inactivated virus stock (1.17x10$^7$ PFU/mL) and viability was calculated at 4hrs and 24hrs.
(TIFF)

**S4 Fig.** **(A)** Representative TUNEL staining in STC exposed to UV-inactivated virus stock (10$^7$ PFU/mL) for 24 hrs. The green fluorescence depicts TUNEL+ cells and shows no cell death **(B)** Human macrophages were exposed to red fluorophore-conjugated E and S1 proteins (1ng) and the uptake was evaluated after 24 hrs by detecting intracellular virus antigens (red) following staining with DAPI (blue) and Phalloidin (green), a marker for actin filaments. High-power magnification pictures depict the uptake of S1 and E by cells while the loss of cytoskeleton was evident only in E-treated cells.
(TIFF)

## Acknowledgments

We thank Ms. Mallory Wilson for her technical help with tissue culture and RT-PCR experiments.

## Author Contributions

**Conceptualization:** Stefanos Giannakopoulos, Daniel P. Strange, Saguna Verma.

**Data curation:** Stefanos Giannakopoulos, Daniel P. Strange, Boonyanudh Jiyarom, Monika A. Ward, Jackson Bakse, Jonathan Yap, Selena Vanapruks, William A. Boisvert, Saguna Verma.

**Formal analysis:** Stefanos Giannakopoulos, Daniel P. Strange, Boonyanudh Jiyarom, Monika A. Ward, Saguna Verma.

**Funding acquisition:** Monika A. Ward, Michelle D. Tallquist, Saguna Verma.

**Investigation:** Stefanos Giannakopoulos, Saguna Verma.

**Methodology:** Stefanos Giannakopoulos, Daniel P. Strange, Boonyanudh Jiyarom, Omar Abdelaal, Aaron W. Bradshaw, Jackson Bakse, Jonathan Yap, Hooman Sadri-Ardekani, Philip Clapp, Sean V. Murphy, Saguna Verma.

**Project administration:** Stefanos Giannakopoulos, Daniel P. Strange, Saguna Verma.

**Resources:** Stefanos Giannakopoulos, Omar Abdelaal, Aaron W. Bradshaw, Vivek R. Nerurkar, Monika A. Ward, William A. Boisvert, Michelle D. Tallquist, Cecilia Shikuma, Hooman Sadri-Ardekani, Philip Clapp, Sean V. Murphy, Saguna Verma.

**Software:** Stefanos Giannakopoulos, Daniel P. Strange, Boonyanudh Jiyarom, Saguna Verma.

**Supervision:** Stefanos Giannakopoulos, Daniel P. Strange, Boonyanudh Jiyarom, Monika A. Ward, Saguna Verma.

**Validation:** Stefanos Giannakopoulos, Daniel P. Strange, Boonyanudh Jiyarom, Monika A. Ward, Jackson Bakse, Jonathan Yap, Saguna Verma.

**Visualization:** Stefanos Giannakopoulos, Daniel P. Strange, Boonyanudh Jiyarom, Monika A. Ward, Jackson Bakse, Jonathan Yap, Saguna Verma.

**Writing – original draft:** Stefanos Giannakopoulos, Saguna Verma.

**Writing – review & editing:** Stefanos Giannakopoulos, Daniel P. Strange, Boonyanudh Jiyarom, Omar Abdelaal, Aaron W. Bradshaw, Vivek R. Nerurkar, Monika A. Ward, Jackson Bakse, Jonathan Yap, Selena Vanapruks, William A. Boisvert, Michelle D. Tallquist, Cecilia Shikuma, Hooman Sadri-Ardekani, Philip Clapp, Sean V. Murphy, Saguna Verma.

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
