## [Decision Letter · Decision Letter 0]

11 Feb 2023

Dear Dr. Verma,

Thank you very much for submitting your manuscript "In vitro evidence against productive SARS-CoV-2 infection of human testicular cells: Bystander effects of infection mediate testicular injury." for consideration at PLOS Pathogens. As with all papers reviewed by the journal, your manuscript was reviewed by members of the editorial board and by several independent reviewers. In light of the reviews (below this email), we would like to invite the resubmission of a significantly-revised version that takes into account the reviewers' comments.

Your manuscript has been reviewed by three experts in the field. All three reviewers thought that the work was of interest, but all three also requested additional data and clarification.

We cannot make any decision about publication until we have seen the revised manuscript and your response to the reviewers' comments. Your revised manuscript is also likely to be sent to reviewers for further evaluation.

Sincerely,

Stanley Perlman

Academic Editor

PLOS Pathogens

Meike Dittmann

Section Editor

PLOS Pathogens

Kasturi Haldar

Editor-in-Chief

PLOS Pathogens

orcid.org/0000-0001-5065-158X

Michael Malim

Editor-in-Chief

PLOS Pathogens

orcid.org/0000-0002-7699-2064

Your manuscript has been reviewed by three experts in the field. All three reviewers thought that the work was of interest, but all three also requested additional data and clarification.

Reviewer's Responses to Questions

**Part I - Summary**

Reviewer #1: It has been hypothesized that elevated immune responses may interpret impaired spermatogenesis in COVID-19 patients. In this manuscript, Giannakopoulos et al. assessed this hypothesis in in vitro primary cell systems and mouse models. They found that SARS-CoV-2 does not productively infect testicular cells. Exposure to basal media from infected airway epithelial cells and plasma from COVID-19 patients decrease the viability and cause death of testicular cells. They demonstrated the SARS-CoV-2 Envelop (E) protein can cause inflammatory responses and cytopathic effects dependent on TLR2. They also observed testicular injury in K18-hACE2 mice after SARS-CoV-2 infection. This research provides evidence to explain the clinical manifestation of testicular symptoms associated with severe COVID-19 and brings insights into the potential mechanism.

Reviewer #2: In the manuscript, Giannakopoulos have investigated the role of SARS-CoV-2 E-protein induced inflammatory response in testis. Authors have used invitro and in vivo system to show that the virus is not productively infecting testicular cells or tissue, but SARS-CoV-2 E protein is sufficient to induce inflammatory response in the system. They further extended data by using supernatant from infected HAE to confirm inflammatory cytokines are sufficient to induce cell death. In general this manuscript is well done and the data supports the overall conclusions of the manuscript but further experiments are required to strengthen the work presented:

Reviewer #3: ‘In vitro evidence against productive SARS-CoV-2 infection of human testicular cells: Bystander effects of infection mediate testicular injury’ by Giannakopoulos et al examines the possibility of SARS-CoV-2 infection of testicular cells and the effect of systemic inflammation on testicular cell viability. The authors convincingly demonstrate that infection of a variety of testicular cell models does not result in an increase in either viral RNA or infectious virions. Given reports of symptoms and pathological findings in autopsy of severe COVID-19 cases, the authors then considered the possibility of other routes of tissue damage. Systemic inflammation, including the hallmark cytokine storm, is observed in severe COVID-19 cases so the authors examined the effect of media from infected HAE cultures and plasma from COVID-19 patients on STC and HTOs. Both sources caused a decrease in viability and an increase in apoptosis in STC and HTOs but not in SC. The authors next considered the role of various SARS-CoV-2 structural proteins in contributing to testicular cell death and demonstrate that exposure to the E protein unique stimulates apoptosis and increased transcription of pro-inflammatory cytokines. Importantly, using the K18 hACE transgenic mouse model, the authors show histopathology in the testes after SARS-CoV-2 infection that is consistent with observed human disease. Overall, these experiments are well performed, however it is unclear that testicular injury is specific to SARS-CoV-2 infection given the very broad mechanism that is shown. Comparison with influenza infection, as an unrelated respiratory virus that also causes systemic inflammation, or similar model is important for the interpretation of the results presented in this manuscript. Questions about the physiologic relevance and biological plausibility of E protein stimulating changes in testicular cells must also be answered. Finally, it is well known that SARS-CoV-2 infection stimulates cytokine and chemokine production including IL-6, TNFa and IL1b and that systemic inflammation can lead to vascular permeability and deleterious effects on a wide variety of cell types and systems which lessens the novelty of this work. A more detailed mechanism would raise the impact of this manuscript.

**Part II – Major Issues: Key Experiments Required for Acceptance**

Reviewer #1: The description “supernatant from HAE” is easy to mistake for the apical wash, especially when the authors did not mention it is from the basal side in figure legends. Authors may change it to basal media or something else. My major concern is the difference between basal media from infected HAE and plasma from COVID-19 patients. Except for the biological differences, the basal media, in this case, contain a large number of viral particles. Did authors find virus in patients’ plasma? Even after UV inactivation, the viral particles may influence cells. This may explain in Figure 4C, after exposure to basal media from infected HAE, TUNEL positive cells are UCHL1 positive cells, while most TUNEL positive cells were not UCHL1 cells after exposure to plasma from COVID-19 patients. Did authors try directly applying UV-inactivated viral particles to STC cells? Another question is about the viral antigen (especially protein E induced immune responses and morphology change). What is the protein level of viral antigens (S, E, N) in plasma from COVID-19 patients or basal media from infected HAE? Could the concentration of E protein be over 1ng/ul? In patients, the antigens are from secreted proteins or viral particles?

Fig1D, except for plaque assay, did authors also check the viral RNA level at an earlier time point (eg. 12hpi)? If the gRNA copy number is higher in +SerP group, it may hint the viral entered cells better in presence of exogenous serine protease, but did not replicate because of the lack of some host factors.

Reviewer #2: 1. The authors have concluded that SARS-CoV-2 E-protein induces inflammation in testis via TLR2 without active virus replication. The data shows that the inflammatory response is specific to E protein and S-protein is not inducing inflammation. Previous report suggest S1 mediated TLR2 activation is sufficient to induce inflammatory response (PMID- 34866574). As in this case S protein failed to induce inflammatory response via TLR2, the disparity in data should be explained.

• The repeat experiments with concurrent exposure to S1, S2, M, N, and E proteins would provide a more comprehensive picture of inflammation in these cells.

2. TLR2 is known to forms a heterodimer with either TLR1 or TLR6 on the cell surface that promotes ligand binding and signal propagation. The basic mechanism of inflammatory response due to E-protein interaction with TLR2 should be investigated.

3. As testis in invivo system are not being infected by SARS-CoV-2, how does the E- protein interacts with the TLR2 on the testicular cells? Is there any evidence of interaction between TLR2 and E protein in invivo system? The biological relevance of this study is not clear.

4. Did author investigate the expression profile of TLR2 in hACE2-K18 mice? Did Authors investigate testis pathology after TLR2 inhibitor administration in these mice? Infection of TLR2 KO/ TLR2 conditional KO mice with mouse adapted virus may provide better insight about the role of TLR2 in testicular inflammation.

5. Figure 7D. TNF and IL-6 shows increased expression at 5dpi. Are these statistically significant?

6. Deleterious effect of inflammation has been well established. In this work, authors have showed deleterious effect of inflammatory cytokines on testicular cells. Why do authors think that the SARS-CoV-2 E protein is major source of damage in testicular cells as compared to systemic inflammation induced due to the direct infection in lungs?

Reviewer #3: The K18 mouse model description is overstated and fails to mention that the severe disease that mice die of is encephalitis and not respiratory. As such, this model is limited in its use for pathogenesis studies. It is important that the authors examine RNA and virus load in the brain in these experiments.

The pathology in Figure 7 is some of the most important data for the conclusions of this manuscript. Is there a scoring scheme/quantitation that can be applied to these samples?

Is this testicular cell injury phenotype unique to SARS-CoV-2 infection? Would media from influenza (or other broadly inflammatory virus) infected cultures have the same effect on testicular cell viability?

**Part III – Minor Issues: Editorial and Data Presentation Modifications**

Reviewer #1: Fig 1B, I believe at MOI=1, all Vero cells must die at 96hpi. Please interpret in figure the dotted line represents the limit of detection (LOD).

In SARS-CoV-2 infected K18-hACE2 mice, authors described testicular edema, germ cell disorganization, and congested tubules were observed in some areas. It is not sure how often this is observed. If they can provide a disease score system to quantify that, it will help readers to have a big picture.

The Fig4C, COVID-19 Plasma pictures are not very convincing for colocalization. Please replace.

Reviewer #2: Minor:

Line 155-156. Reference should be added.

Figure 2D Statistical analysis should be shown.

SARS-CoV-2 E-protein generation and purification should be mentioned in Method section

Reviewer #3: If exposure to E protein alone is enough to induce cytokine secretion (Fig 5D-G), why was exposure to virions not able to induce the same response (Fig 1E)? Is there any evidence of circulating E protein in the blood of COVID-19 patients? How would a transmembrane protein be stable in circulation?

Figure 2- Similar data has been previously published and this would be more appropriate as a supplemental figure

PLOS authors have the option to publish the peer review history of their article (what does this mean?). If published, this will include your full peer review and any attached files.

Reviewer #1: No

Reviewer #2: **Yes: **Abhishek Verma

Reviewer #3: No
---

## [Editor Report · Decision Letter 1]

8 May 2023

Dear Dr. Verma,

We are pleased to inform you that your manuscript 'In vitro evidence against productive SARS-CoV-2 infection of human testicular cells: Bystander effects of infection mediate testicular injury.' has been provisionally accepted for publication in PLOS Pathogens.

Best regards,

Stanley Perlman

Academic Editor

PLOS Pathogens

Meike Dittmann

Section Editor

PLOS Pathogens

Kasturi Haldar

Editor-in-Chief

PLOS Pathogens

orcid.org/0000-0001-5065-158X

Michael Malim

Editor-in-Chief

PLOS Pathogens

orcid.org/0000-0002-7699-2064

Your manuscript is basically acceptable for publication. Please review it for errors in writing. For example, the title of Figure legend 1 does not appear to be correct.
---

## [Editor Report · Acceptance letter]

15 May 2023

Dear Dr. Verma,

We are delighted to inform you that your manuscript, "In vitro evidence against productive SARS-CoV-2 infection of human testicular cells: Bystander effects of infection mediate testicular injury.," has been formally accepted for publication in PLOS Pathogens.

Best regards,

Kasturi Haldar

Editor-in-Chief

PLOS Pathogens

orcid.org/0000-0001-5065-158X

Michael Malim

Editor-in-Chief

PLOS Pathogens

orcid.org/0000-0002-7699-2064